# Molecular Dynamic Simulation Analysis of a Novel Missense Variant in *CYB5R3* Gene in Patients with Methemoglobinemia

**DOI:** 10.3390/medicina59020379

**Published:** 2023-02-16

**Authors:** Asmat Ullah, Abid Ali Shah, Fibhaa Syed, Arif Mahmood, Hassan Ur Rehman, Beenish Khurshid, Abdus Samad, Wasim Ahmad, Sulman Basit

**Affiliations:** 1Department of Biochemistry, Faculty of Biological Sciences, Quaid-i-Azam University, Islamabad 45320, Pakistan; 2Center for Medical Genetics & Hunan Key Laboratory of Medical Genetics, School of Life Sciences, Central South University, Changsha 410083, China; 3Department of General Medicine, Pakistan Institute of Medical Sciences, Shaheed Zulfiqar Ali Bhutto Medical University, Islamabad 44000, Pakistan; 4Department of Endocrine and Diabetes, Cholchester General Hospital, Cholchester CO4 5JL, UK; 5Department of Biochemistry, Abdul Wali Khan University, Mardan 23200, Pakistan; 6Department of Biochemistry and Molecular Medicine, College of Medicine, Taibah University, Medina 42318, Saudi Arabia; 7Center for Genetics and Inherited Diseases, Taibah University, Medina 42318, Saudi Arabia

**Keywords:** recessive congenital methemoglobinemia (RCM), exome sequencing, *CYB5R3*, molecular dynamics simulation

## Abstract

*Background and Objective*: Mutations in the *CYB5R3* gene cause reduced NADH-dependent cytochrome b5 reductase enzyme function and consequently lead to recessive congenital methemoglobinemia (RCM). RCM exists as RCM type I (RCM1) and RCM type II (RCM2). RCM1 leads to higher methemoglobin levels causing only cyanosis, while in RCM2, neurological complications are also present along with cyanosis. *Materials and Methods*: In the current study, a consanguineous Pakistani family with three individuals showing clinical manifestations of cyanosis, chest pain radiating to the left arm, dyspnea, orthopnea, and hemoptysis was studied. Following clinical assessment, a search for the causative gene was performed using whole exome sequencing (WES) and Sanger sequencing. Various variant effect prediction tools and ACMG criteria were applied to interpret the pathogenicity of the prioritized variants. Molecular dynamic simulation studies of wild and mutant systems were performed to determine the stability of the mutant CYB5R3 protein. *Results*: Data analysis of WES revealed a novel homozygous missense variant NM_001171660.2: c.670A > T: NP_001165131.1: p.(Ile224Phe) in exon 8 of the *CYB5R3* gene located on chromosome 22q13.2. Sanger sequencing validated the segregation of the identified variant with the disease phenotype within the family. Bioinformatics prediction tools and ACMG guidelines predicted the identified variant p.(Ile224Phe) as disease-causing and likely pathogenic, respectively. Molecular dynamics study revealed that the variant p.(Ile224Phe) in the CYB5R3 resides in the NADH domain of the protein, the aberrant function of which is detrimental. *Conclusions*: The present study expanded the variant spectrum of the *CYB5R3* gene. This will facilitate genetic counselling of the same and other similar families carrying mutations in the *CYB5R3* gene.

## 1. Introduction

The gene *CYB5R3* (MIM#613213) translates into cytochrome b5 reductase enzyme. The mutated enzyme causes recessive congenital methemoglobinemia (RCM, MIM#250800) characterized by the low availability of oxygen in blood cells. Under normal conditions, the CYB5R3 enzyme in erythrocytes converts the methemoglobin to its ferrous state (Fe^++^); however, due to genetic mutations in the *CYB5R3* gene, it cannot be converted back into ferrous state (Fe^++^) from ferric (Fe^+++^) state, resulting in methemoglobinemia. Furthermore, in this medical condition, the methemoglobin level is greater than 1% and ferrous iron (Fe^++^) is oxidized to the ferric state (Fe^+++^), culminating in tissue hypoxia [1,2].

Methemoglobinemia can occur both as congenital and acquired conditions. Congenital methemoglobinemia type I is rare and caused due to *CYB5R3* defects. On the other hand, acquired methemoglobinemia, much more common than the congenital form, occurs due to exposure to certain chemical substances. These substances include direct oxidizing agents (benzocaine and prilocaine), indirect oxidation (nitrates), or metabolic activation (e.g., aniline and dapsone) [3], which lead to the oxidation of hemoglobin and ultimately form methemoglobin in higher amounts. Consequently, the hemoglobin is unable to convert the iron back to its ferrous state [4]. There are also mutations in the globin genes, forming various variants of hemoglobin (MetHb), stabilizing the iron in Fe^+++^ state [5].

Cytochrome b5 reductase enzyme exists in two forms: soluble and membrane-bound forms. The soluble form is 31 kDa, located in the cytosol of RBCs which reduce MetHb (methemoglobin) to functional Hb (hemoglobin). The other membrane-bound form is a 34 kDa attached to the ER, plasma membrane, and outer mitochondrial membrane through a 14 carbon fatty acid, myristoyl group in somatic cells and participates in notable activities such as fatty acid/cholesterol biosynthesis and cytochrome P450 pharmacokinetics [6,7,8].

The *CYB5R3* gene encoding cytochrome-b5 reductase 3 is involved in cellular redox and metabolic hemostasis [5]. Recently, numerous studies have determined the molecular basis of patients with methemoglobinemia. Pathogenic variants in the CYB5R3 enzyme cause two types of methemoglobinemia. The most common methemoglobinemia is type 1 which includes a tolerable form marked by a highly documented clinical representation of cyanosis. Methemoglobinemia type II (generalized) affects all tissues of the body and manifests in cyanosis as well as neurological complications [9,10]. Recently, congenital methemoglobinemia was also reported in a patient with multiple limb anomalies [11]. Similarly, in another study, the authors showed that loss of function mutations affecting both the FAD- and NAD-binding domains in *CYB5R3* gene resulted in both type I and type II recessive congenital methemoglobinemia [12].

To decipher the complex interplay of the clinical and genetic interactions of recessive congenital methemoglobinemia, the use of whole exome sequencing technologies have accelerated the molecular diagnosis significantly. To identify the etiology of patients with RCM1, we aim to identify disease-causing variants in the *CYB5R3* gene in a family of Pakistani origin. Further, the CYB5R3 wild type (CYB5R3^WT^) and mutant (CYB5R3^MT^) were subjected to molecular dynamics simulation to ascertain the behavior of the two proteins in terms of protein stability, flexibility, and compactness.

## 2. Materials and Methods

### 2.1. Ethical Approval and Study Subjects

Approval of the study was provided by the Institutional Review Board (IRB) of Quaid-i-Azam University, Islamabad, Pakistan. We recruited a family showing a congenital metabolic disorder from a remote region of the country, and the members were clinically characterized at the Pakistan Institute of Medical Sciences (PIMS), Islamabad, Pakistan. A pedigree was drawn after obtaining the information from older members of the family. Informed written consent was obtained from all individuals who participated in the study.

### 2.2. Blood Sampling and Extraction of Genomic DNA

For the genetic study, 3–5 mL blood samples were collected from all available healthy and affected individuals of the family. DNA was extracted from the blood using a commercially available DNA extraction kit (QIAGEN, Germantown, MD, USA) following standard protocols.

### 2.3. Whole Exome Sequencing

Whole exome sequencing was performed using the DNA of two affected individuals (IV-1 and IV-2). Exome of the patients was sequenced using the Illumina platform (Illumina HiSeq 2500 sequencer) after preparing libraries using Agilent SureSelect Target Enrichment Kit following the manufacturer’s instructions. Read calling, alignment, and annotation of variants were performed using the BWA Enrichment application of BaseSpace, Burrows–Wheeler Aligner (BWA), and Illumina Variant Studio v2.2, respectively, as described previously by [13].

To identify pathogenic variants in the exomes of the patients, variants having a minor allele frequency of more than 0.01 in any of the human genome population databases including 1000 Genomes (https://www.internationalgenome.org/ accessed on 14 August 2022), ExAC (https://ExAC.broadinstitute.org/ accessed on 14 August 2022), gnomAD (https://gnomad.broadinstitute.org/ accessed on 14 August 2022), Sequencing Project 6500 (ESP6500) (https://evs.gs.washington.edu/EVS/ accessed on 14 August 2022) were filtered out. Synonymous variants were filtered out from the list of prioritized variants in each of the two exomes. Variants shared by the exomes of both siblings were selected for further analysis. The selected variants were divided into two groups of heterozygous and homozygous variants. The list of homozygous variants was analyzed for homozygous pathogenic variants, while the list of heterozygous variants was analyzed for the identification of heterozygous variants that might have segregated as a compound heterozygous or digenic inheritance in affected individuals. Based on the inheritance pattern, and consanguinity, homozygous variants were prioritized. Variants were associated with the disease phenotypes based on the expression and function of the gene in the cellular metabolism and/or the previous association of pathogenic variants in the same gene with similar phenotypes. To exclude the polymorphic nature of the identified variants, 183 ethnically matched control exomes were analyzed.

### 2.4. Sanger Sequencing

To assess the segregation of the variants with the disease phenotypes, DNA from all the available normal and affected individuals was Sanger sequenced. Primers were designed using Primer3 software (https://primer3.ut.ee/ accessed on 14 August 2022). Hits of the designed primers were assessed using Ensembl BLAST/BLAT tool (https://www.ensembl.org/Multi/Tools/Blast accessed on 14 August 2022). In silico PCR using primers having a single hit was performed on the UCSC In-Silico PCR tool (https://genome.ucsc.edu/cgi-bin/hgPcr accessed on 14 August 2022). The exon containing the identified variant was amplified using exon-specific primers through polymerase chain reaction following standard protocols. Sanger sequencing was performed on ABI Prism 310 Genetic Analyzer (Applera, Foster City, CA, USA) using the BigDye Terminator v3.1 Cycle Sequencing Kit. Sanger sequencing data was analyzed using BIOEDIT sequence alignment editor, version 7.2 (https://bioedit.software.informer.com/7.2/ accessed on 14 August 2022). Reference sequence of *CYB5R3* was downloaded from the Ensembl genome browser (https://www.ensembl.org/index.html accessed on 14 August 2022).

### 2.5. Determination of Variant Pathogenicity in CYB5R3

Various in silico tools were used to predict the pathogenicity of the identified missense variant in *CYB5R3*. These included SIFT [14], PolyPhen2 [15], LRT [16], MutationTaster [17], MutationAssessor [18], PROVEAN [19], VEST [20], FATHMM [21], CADD [22], Genocanyon [23], REVEL [24], VarSome [25] and ClinPred [26]. ACMG guidelines were applied to interpret the identified variant [27]. The frequency of the mutant allele was examined using population databases such as gnomAD [28], and 1000 Genomes [29]. Conservation of the mutated amino acid across distinct species was checked online using Homologene (https://www.ncbinlm.nih.gov/homologene accessed on 14 August 2022).

### 2.6. Data Retrieval and Structure Preparation

The crystal structure of cytochrome b (5) reductase (PDB ID: 1UMK) was retrieved from the PDB database [1]. The crystal water particle was removed and the 3D structure was subjected to loop modeling in order to correct the missing residues, followed by energy minimization with the MMFF94s force field implemented in molecular operating environment (MOE). The mutation was induced in the structure using the mutagenesis tool implemented in PyMol software [2]. The structure of heme was retrieved from PubChem database, and subjected to optimization using DFT/B3LYP/3-21G framework implemented in Gaussian 09 software package [3]. The optimized structure was docked against the receptor in both states, i.e., mutant and wild. The best-docked complexes were subjected to a total of 100 ns MD simulations in triplicate.

### 2.7. Molecular Dynamics Simulation

We carried out molecular dynamics (MD) simulations and analyses using AMBER v2021 with the force field (ff14SB) [30]. The MCPB.py python script was used for the heme parameterization [4]. To neutralize the system, counter ions (Na^+^ and Cl^−^) were added using the LEAP module. The octahedral box of TIP3P water model with a 12.0 Å buffer was used for solvating the whole system [31]. For the van der Waals and long-range electrostatic interactions, a cutoff distance of 10 Å was used. For treatment of long-range electrostatic interactions, the particle mesh Ewald (PME) algorithm was used. The SHAKE algorithm was used to constrain the bonds involving hydrogen atoms, followed by 0.5 nanosecond (ns) of constant pressure equilibration at 300 K [32]. For controlling the temperature, Langevin dynamics was used [33]. Finally, 100 ns MD simulation in triplicate was carried out with the CUDA version of PMEMD in GPU for all of the equilibrated complex systems at constant temperature and pressure [34].

### 2.8. Post-Simulation Analysis

Using CPPTRAJ module of Amber20 software, we attempted to determine the root mean square deviation (RMSD), root mean square fluctuation (RMSF), and radius of gyration (Rg) for wild-type and mutant Ile224Phe system. PyMol and Origin were used for visualization of structure and graphical representations [35].

## 3. Results

### 3.1. Clinical Features

In the present study, a family with one female (IV-1) and one male (IV-2) affected individual was clinically examined and genetically analyzed. Patient IV-1 was a 16-year old girl; who had had bluish discoloration of her peripheries for 2 months, which was aggravated with exertion, chest pain radiating to her left arm, New York Heart Association (NYHA) grade II dyspnea, breathlessness and on and off coughing up of blood from the lungs (hemoptysis). Upon systemic evaluation, undocumented fever, night sweats, and undocumented weight loss were also noted. On physical examination, she had blood pressure (BP) of 120/70 (normal 120/80), heart rate of 125/min (60–100/min), respiration rate (RR) of 15/min (12–16/min), and saturation of 86% on room air along with peripheral and central cyanosis, while the rest of her examination was unremarkable. Her arterial blood gases showed decreased PO_2_ level of 32 mmHg (normal 75–100 mmHg) while CO_2_, pH, and bicarbonate levels were normal. Complete blood count (CBC) showed a hemoglobin level of 15.8 g/dL (normal 11.6–15 g/dL for women) with normal erythrocyte morphology and indices, leukocytes count of 8.57 × 10^9^/L (normal 4.5–11.0 × 10^9^/L) within normal range, thrombocytes count of 241 × 10^9^ (normal 150 to 400 × 10^9^/L), reticulocyte count of 3.8% (normal 0.5–2.5%), red cell distribution width (RDW) of 47.1 fL (normal 39–46 fL). Liver and renal function tests as well as serum electrolytes were within an acceptable range. Her chest X-ray and echocardiogram were also insignificant. On further examination, her hemoglobin studies showed bands of HbA, HbA2, and MetHb. HbF was 1.2% (normal 0.8% to 2%), HbA2 was in the range of 3.3% (normal 2.0–3.5%), while MetHb (normal 1–2%) level was 49%, pointing towards congenital methemoglobinemia.

The affected subject (IV-2) is a 17-year old male having bluish discoloration of fingers, toes, lips, and tongue. He also had exertional dyspnea and venesection twice for polycythemia. On physical examination, he had blood pressure (BP) of 110/90 (120/80), heart rate of 115/min (60–100/min), respiratory rate (RR) of 13/min (12–16/min) and saturation of 80% on room air along with peripheral and central cyanosis, while the rest of the examination was unremarkable. His arterial blood gases showed a decreased PO_2_ level of 44 (normal 75–100 mmHg) while CO_2_, pH, and bicarbonate were normal. His CBC examination showed hemoglobin level of 17.0 g/dL (normal 13.2–16.6 g/dL for men) with normal red cell morphology and indices, white cell count of 10.11 × 10^9^/L (normal 4.5−11.0 × 10^9^/L) with normal differential count, platelet count of 241 × 10^9^ (normal 150−400 × 10^9^/L), reticulocyte count of 3.4% (normal 0.5–2.5%) and red blood cell distribution width of 49.3 fL (normal 39–46 fL). His liver and renal function tests as well as serum electrolytes were normal. His chest X-ray and echocardiogram were normal. Hemoglobin investigation showed bands of HbA, HbA2, and MetHb. HbF was 1.5% (normal 0.8–2%), HbA2 as 3.1% (normal 2.0–3.5%), while MetHb was 50.5% manifesting congenital methemoglobinemia (Table 1). The third affected individual (IV-5) was not available for clinical examination, but shared his blood for genetic analysis.

### 3.2. Molecular Genetic Analysis

The pedigree of the present family showed consanguinity between the parents of the patients and an autosomal recessive inheritance pattern of the disease (Figure 1A). Whole exome sequencing revealed a novel homozygous missense variant (NM_001171660): c.670A > T: NP_001165131.1: p.(Ile224Phe) in *CYB5R3* in the affected individuals. Their parents were heterozygous carriers for the variant (Figure 1B). The variant detected is absent from population databases such as gnomAD, 1000 Genomes, ESP6500, and 183 ethnically matched control exomes. The variant p.(Ile224Phe) was predicted to be pathogenic by various in silico variant effect prediction tools (Appendix A). Moreover, the amino acid Ile224 was found conserved across several different species (Figure 1C). According to ACMG classification, the variant was interpreted as likely pathogenic according to the following criteria; PM1: Located in a mutational hot spot and/or critical and well-established functional domain (e.g., active site of an enzyme) without benign variation, PM2: Absent from controls (or an extremely low frequency if recessive) in Exome Sequencing Project, 1000 Genomes Project, or Exome Aggregation Consortium, PP3: Multiple lines of computational evidence support a deleterious effect on the gene or gene product (conservation, evolutionary, splicing impact, etc., and PP4: Patient’s phenotype or family history is highly specific for a disease with a single genetic etiology, respectively.

### 3.3. Heme Docking and Predicted Interactions with Native and Mutant CYB5R3

The heme docking studies with the native and mutant CYB5R3 showed a wide range of potential interactions (Figure 2E). Binding docking tools of MOE with MMFF94 force field were used to analyze the interactions of optimized heme group with mutant as well as native protein receptor. A total of 20 poses were generated for both the systems based on the docking scores. The best binding pose was selected for the analysis of the interactions and MD studies. The interaction details and docking score are listed in Table 2.

Heme established a significant number of hydrogen bond donor and acceptor interactions with the binding site residues of native CYB5R3. It was observed that in the case of wild-type Cys273, establishing two hydrogen donor and one hydrogen bond acceptor interactions, Thr181 contributes to one hydrogen bond donor interaction, while His117, Phe120, Tyr112 form H–pi and pi–pi interaction with the heme group. The interaction analysis of the mutant state of CYB5R3 reveals that Pro275, Pro276, and Tyr112 form one hydrogen bond acceptor with a docking score of −7.89. The findings of docking exposed that the point mutation at position 224 could significantly reduce the interaction of heme toward the CYB5R3. To gain deeper insights into the conformational changes inserted by point mutation, the docked complexes were subjected to 100 ns of MD simulations in triplicate.

### 3.4. Molecular Dynamics Simulations

A 100 ns MD simulation run in triplicate was performed on both CYB5R3 ^WT and MT^ systems to monitor the stability using RMSD (root mean square deviation), RMSF (root mean square fluctuation), Rg (radius of gyration), using the MD simulation trajectories via Schrödinger 2021-2.

A lower RMSD curve during dynamic simulation translates into lower protein stability. As shown in Figure 2B, the RMSD values of both the native and mutant CYB5R3 showed the same trajectory till 20 Å, after which the mutant demonstrated greater deviation from the native protein in all triplicates. We can conclude that mutant p.(Ile224Phe) retained maximum deviation for the greater part of the simulation than wild CYB5R3, thus affecting the stability of the CYB5R3 protein.

To seek additional information about the effect of the mutation on protein structure and function, the per-residue fluctuation was computed. In all three runs, as shown in Figure 2C, the mutant system showed an increased RMSF trajectory and, hence, greater flexibility while the wild type depicted lower RMSF and lesser flexibility. In addition, Rg was used to define the compactness of wild and mutant proteins. It is only a time-based measurement of the distance between the mass of the protein center and its terminus. Normally, globular proteins have less gyration, while extended proteins have a higher Rg value. The radius of gyration of wild CYB5R3 showed an increased trend till the end of the simulation in all three runs, shown in Figure 2D. So, wild type CYB5R3 has greater Rg and is less tightly packed than mutant CYB5R3 which is more tightly packed and has less Rg.

## 4. Discussion

NADH-cytochrome b5 reductase 3 deficiency is an important genetic cause of autosomal recessive congenital methemoglobinemia (RCM). An important substrate of the CYB5R3 protein is CYBR, which plays a crucial role in the electron-transfer reactions performed by CYB5R3. CYB5 is a heme-containing protein expressed in animals, plants and fungi that acts as an electron transporter in a variety of important reactions [36,37]. Each heme group has iron in the reduced (Fe^+2^) state. When iron reacts with oxygen (O_2_), it transfers an electron to make oxyhemoglobin. Upon reaching the tissues, this iron returns back to its original Fe^+2^ state. If mishaps occur, such as oxidative stress or any loss during oxygen delivery, it fails to attach an electron to combine with O_2_ and the iron remains in a Fe^+3^ state. So, when the iron is in Fe^+3^ state or meth form, it is not capable of transporting the O_2_ molecule and hence disease manifests itself [38]. The sequence of this transfer is summarized as follows:

2^e^- from NADH→CyB5R→FAD→ reduction of 2 CyB5→electron transfer to MetHb.

The flavin reduction step has been shown to be the rate-limiting factor of this chain reaction [39].

CYB5R3 and the substrate CYB5 are widely expressed proteins, which mainly perform the function of NADH-dependent electron transport, where CYB5R3 transfers two reducing equivalents from NADH to FAD seated in the FAD-binding domain, and lastly to CYB5 [40].

It should be noted that reducing MetHb is contingent on the oxidoreductase enzymatic properties of cb5r which needs reduced NADH and NADPH as substrate molecules. NADH is made during the Embden–Meyerhof pathway while NADPH in the pentose phosphate pathway (PPP) of RBCs. Both of these substrate molecules reduce cb5, which sequentially reduces the oxidized Fe^+3^ of Hb, when an electron is transferred from reduced cb5 [41].

Over the years, studies have reported methemoglobinemia known as RCM1, in patients positive for cyanosis without neurological complications due to *CYB5R3* genetic aberrations [42]. With time, numerous studies have reported type I methemoglobinemia with documented clinical features of cyanosis and other associated symptoms of fatigue, weakness, headache, metabolic acidosis due to mutations in cytochrome b5 reductase [4,43,44,45].

In the present study, we identified a novel homozygous missense variant p.(Ile224Phe) in the *CYB5R3* gene, located on chromosome 22q13.2, segregating with RCM1 in a Pakistani family.

Until now, 80 homozygous and compound heterozygous genetic variants have been documented in the *CYB5R3* in the pathogenic course of methemoglobinemia among different ethnic groups around the world (http://www.hgmd.cf.ac.uk/ac/all.php accessed on 14 August 2022).

A molecular study of RCM1 in a Japanese patient uncovered a homozygous missense variant p.(Lys149Pro) in the *CYB5R3* gene [42]. Later on, Higasa et al. (1998) analyzed the *CYB5R3* gene in two patients originating from Thailand and found two novel homozygous variations including a missense variant causing RCM1 and a nonsense variant leading to RCM2. Thus, the authors concluded that the nonsense variant in *CYB5R3* led to a severe form of methemoglobinemia because of the malfunction of the truncated protein, while the missense variant caused RCM1, due to degradation of the unstable mutant enzyme with normal activities in the patient’s erythrocytes [46].

The primary symptom in the patients under study was cyanosis. The neurological phenotypes were not observed in our present cases. Thus, based on a clinical profile, the clinician diagnosed the present cases as RCM1. The genetic study validated the diagnosis by the identification of a homozygous missense variant in *CYB5R3* in the patients. In addition to peripheral and central cyanosis, one of the patients in the family studied in this work complained of chest pain radiating to the left arm, dyspnea, and hemoptysis, common symptoms of RCM. The rest of the metabolic tests such as a complete blood count, liver and renal function tests and serum electrolytes were normal. Imaging studies including chest X-ray were unremarkable, and the echocardiogram was also insignificant.

The CYB5R3 protein is composed of two discrete domains: The N-terminal FAD binding domain (Ile34-Arg143), which contains a binding site for the FAD prosthetic group, and the NADH domain (Lys173-Phe301). At the core of the protein is a large interdomain cleft (Gly144-Val172) known as a hinge region [47]. The residue isoleucine, mutated in the present study (p. (Ile224Phe)), resides in the NADH domain (Pfam: PF00175) and is conserved across different species.

Molecular dynamics simulation is an effective method to observe the bona fide behaviors of various molecules along with their surroundings [48]. In our study, we carried out molecular dynamic simulations on two systems, native and CYB5R3^p.Ile224Phe^. We studied RMSD, RMSF and Rg analysis between the native and mutant (Ile224Phe) CYB5R3 protein structures. In order to avoid random conclusions, we conducted triplicate MD simulations for all the systems. In terms of RMSD, mutant CYB5R3 showed maximum deviation for the greater part of the simulation as shown by the higher RMSD curve than the wild CYB5R3. The RMSF trajectory showed an increased flexibility for the mutant (Ile224Phe) CYB5R3 compared to the native protein. Rg analysis, which illustrates the overall spread of a molecule, indicated greater Rg and less compaction for wild type CYB5R3 than mutant CYB5R3 which was more tightly packed and had less Rg.

Although we have uncovered a novel missense variant through the WES approach and have provided in silico evidence for the variant, the main caveat in our study is still the functional validation of this missense variant in CYB5R3 gene using traditional in vitro and in vivo approaches.

## 5. Conclusions

Through WES, we identified a likely pathogenic novel homozygous missense variant in the *CYB5R3* gene in a family of Pakistani origin, by segregating RCM1 in an autosomal recessive manner. Docking analysis depicted the native and mutant CYB5R3 potential interactions. Molecular dynamics simulation showed that the missense variant in *CYB5R3* leads to instability of the mutant system in comparison to wild-type protein as elucidated by the different trajectories of RMSD, RMSF and Rg.

To the best of our knowledge, this is the first case of RCM1 from the Pakistani population. Therefore, the study will be helpful for genetic counseling, carrier testing, and bringing awareness regarding genetic disorders caused by consanguinity to the population. The study also expands the variant spectrum of *CYB5R3* and contributes to the evaluation of genotype–phenotype correlation.

## Figures and Tables

**Figure 1 medicina-59-00379-f001:**
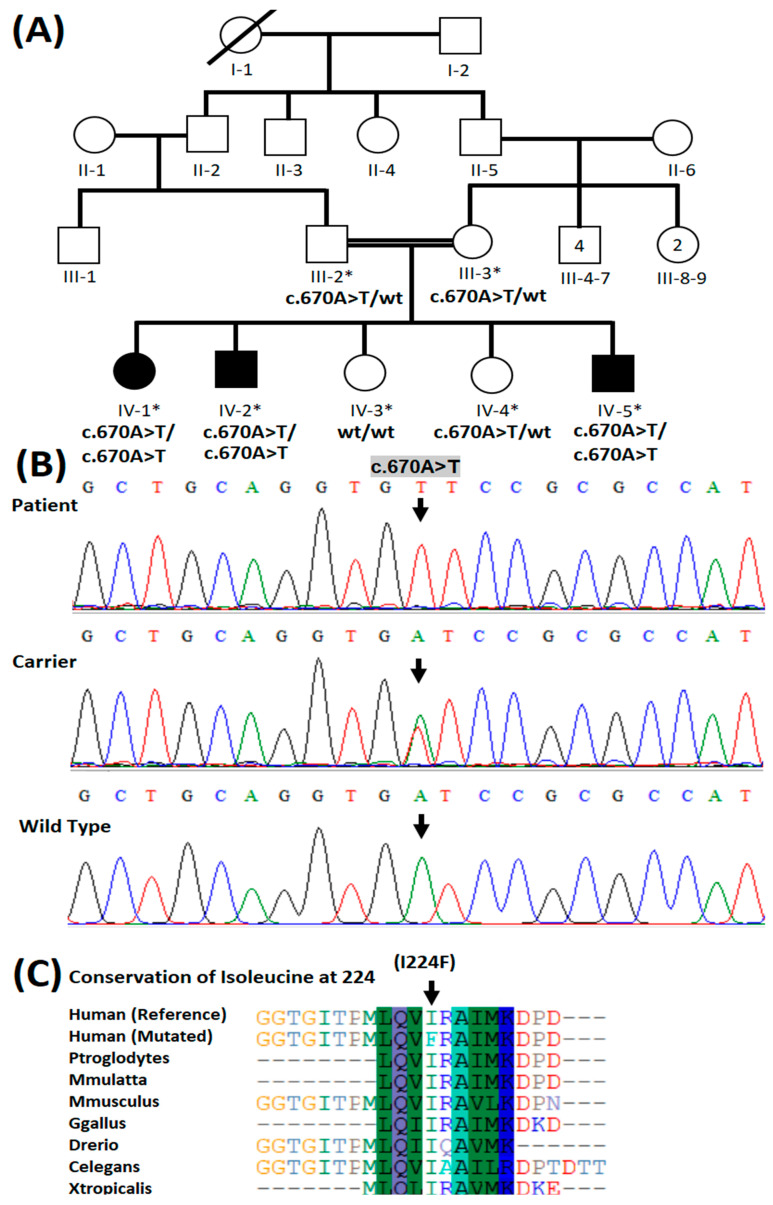
(**A**) Four-generation pedigree of a family showing consanguinity and autosomal recessive mode of inheritance of the disease. Squares represent males while females are denoted by circles. In this family pedigree, filled squares and circles indicate the presence of methemoglobinemia in male and female, and white squares and circles represent healthy males and females, respectively. The asterisk symbol above individuals represents those who participated in the study. Genotypes have been shown beneath each individual participating in the present study. (**B**) Partial sequence of exon 8 of *CYB5R3* showing variant c.670A > T in homozygous form in patients, heterozygous form in carriers, and wild type in healthy control individuals. An arrow indicates the site of a variant in the patient. (**C**) Conservation of isoleucine at amino acid position 224 in CYB5R3 protein in species with a known CYB5R3 ortholog.

**Figure 2 medicina-59-00379-f002:**
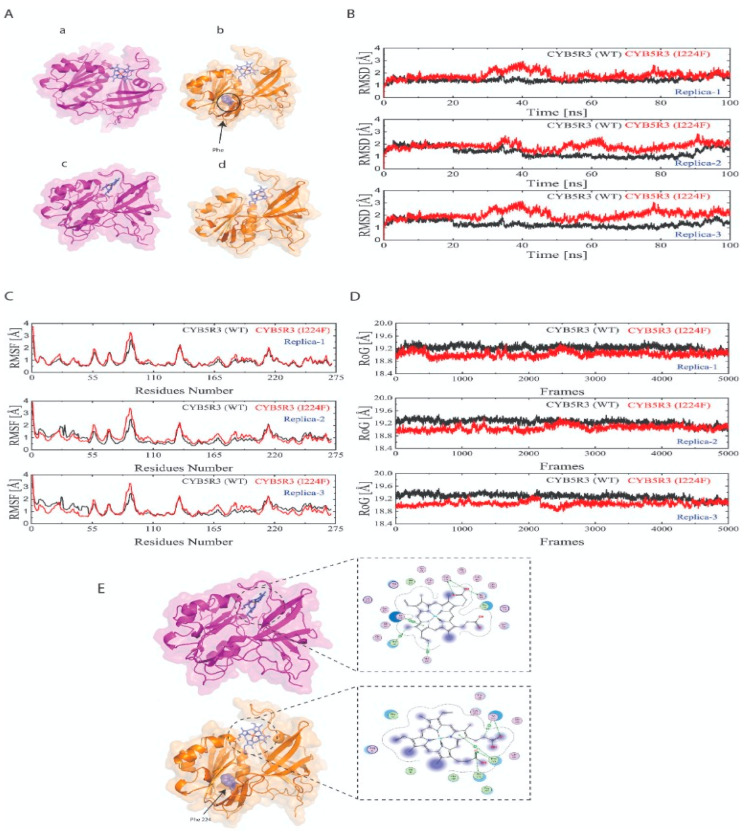
CYB5R3 before (a,b) and after (c,d) molecular dynamic simulations (**A**). Molecular dynamics simulation of the CYB5R3^WT^ (black curve) and ^MT^ (red curve) generated over 100 ns in triplicate showing the root mean square deviation (RMSD) (**B**); root mean square fluctuation (residual fluctuations over the simulation time) (**C**); radius of gyration (**D**). Heme docking with the native and mutant CYB5R3 showing potential interactions (**E**).

**Table 1 medicina-59-00379-t001:** Hematological, biochemical, and genetic findings in affected individuals with recessive congenital methemoglobinemia type 1, reported in the present study.

Parameter	Patient 1 (IV-I)	Patient 2 (IV-2)	Reference Value
Age	16	17	
Gender	F	M	
Consanguinity	Yes	Yes	
**Hematological and biochemical analysis**			
Cyanosis	Yes +	+	
Dyspnea	+	+	
Orthopnea	+	−	
Hemoptysis	+	−	
Blood pressure	120/70	110/90	120/80
Heart rate (min)	125	115	120/min
R.R (min)	15	13	12–16/min
Arterial PO_2_	32	44	75–100 mmHg
PH, CO_2_, HCO_3_^−^	N	N	
Hemoglobin (g/dL)	15.8	17	13.2–16.6 g/dL (men), 11.6–15 g/dL (women)
Leukocytes (g/dL)	8.57 × 10^9^	10.1 × 10^9^	4.5–11.0 × 10^9^ g/dL
Thrombocytes/L	241 × 10^9^	241 × 10^9^	150–400 × 10^9^/L
Reticulocyte Count (%)	3.80%	3.40%	0.5–2.5%
Red cell distribution width (fL)	47.1 fL	49.3 fL	39–46 fL
Blood cell count	N	N	
Liver function test	N	N	
Renal function test	N	N	
Serum electrolytes	N	N	
Pulmonary tuberculosis	−	−	
Hb investigations			
HbA	−	−	
HBA2	3.30%	3.10%	2.0–3.5%
MetHb	49%	50.50%	1–2%
HbF	1.20%	1.50%	0.8–2%
Recessive congenital methemoglobinemia	RCM Type I	RCM Type I	
Genetic analysis	*CYB5R3*; p.(Ile224Phe)	*CYB5R3*; p.(Ile224Phe)	

**Table 2 medicina-59-00379-t002:** Represents the docking scores and reports of predicted interactions of heme toward native and mutant CYB5R3.

Protein	Interacting Residues	Interaction Type	Distance	Energy (Kcal/mol)	Docking Score
Native	CYS 273	H-donor	3.10	−0.3	−8.34
	THR 181	H-donor	2.99	−2.5	
	CYS 273	H-donor	3.75	−0.2	
	CYS 273	H-acceptor	3.10	−0.6	
	HIS 117	H-pi	4.35	−0.6	
	PHE 120	H-pi	4.84	−0.1	
	TYR 112	pi-pi	3.96	−0.0	
Mutant	PRO 275	H-acceptor	3.70	−0.1	−7.89
	PRO 276	H-acceptor	3.66	−0.2	
	TYR 112	H-acceptor	3.36	−0.1	
	TYR 112	H-pi	4.50	−0.1	
	PRO 275	pi-H	4.72	−0.1	

## Data Availability

Data available on request from the authors.

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
