# Peer review of "Molecular Dynamic Simulation Analysis of a Novel Missense Variant in CYB5R3 Gene in Patients with Methemoglobinemia"

_medicina, 2023, doi:10.3390/medicina59020379_

Round 1

Reviewer 1 Report (New Reviewer)

The authors investigate several members of the same family with methemoglobinemia and uncover a novel missense mutation in the CYB5R3 gene using whole exon sequencing.

It is not clear the benefits of the WES approach since laboratory investigations detected the presence of Hb M. Is this correct? Hb Ms are a specific group of Hb variants that have a perturbed heme pocket that allows the iron moiety to be oxidised. Do the authors mean methaemoglobin (MetHb)?

Since MetHb was detected, a targeted screen would make sense as this would reduce the need for bioinformatics. Can the authors explain their approach?

The Introduction could be more concise while the discussion would benefit from being more focussed on Type I congenital methemoglobinemia as it is relevant to the findings of the investigation. There is also an issue with formatting of the text as part of the results and the Discussion has merged with Figure 2 legend.

Introduction lines 75 to 77 the authors state that the Type II RCM variants affect the FAD binding domain while Type I RCM are associated with the NAD binding domain and refer to  reference 13. Figure 3 from this reference shows that both types of variants are present in both regions of the protein. This is too simplified and is related more to the combination of the mutations, whether they cause a truncated protein etc resulting in the total activity of cytochrome b5 reductase.

Extensive variant classification was performed and presented in parts D and E of Figure 1. The resolution makes it difficult re read. If space is limited then place in a supplementary section so the results of various prediction tools can be analysed.

The term ACMG criteria requires explanation. What criteria did the authors use to define the variant as likely pathogenic? List the criteria used for the variant classification.

There are errors of grammar and syntax. The paper would benefit from review by a person whose first language is English and who is familiar with the terminology.

Author Response

Response to Reviewer 1

Thank you for your efforts in reviewing this manuscript. Your comments are very much practical and applicable which have improved the quality of this research. We are happy to write that we have addressed your concerns below. We look forward to receiving a positive response from you.

Comment 1:

It is not clear the benefits of the WES approach since laboratory investigations detected the presence of Hb M. Is this correct? Hb Ms are a specific group of Hb variants that have a perturbed heme pocket that allows the iron moiety to be oxidized. Do the authors mean methemoglobin (MetHb)?

Answer: Thanks for mentioning the concern related to WES approach. The present family was a suspected case of methemoglobinemia based on clinical profile. We were interested to know the genetic cause of the disease phenotypes to help the affected family in carrier screening, genetic counseling and prenatal diagnosis. After the identification of the homozygous variant in CYB5R3 through WES, screening of phenotypically healthy siblings revealed the heterozygous status of IV-4. These findings and predicted consequences of cousin marriages were shared with the affected family during genetic counseling.

Comment 2:

Since MetHb was detected, a targeted screen would make sense as this would reduce the need for bioinformatics. Can the authors explain their approach?

Answer: We understand your concern, but, in populations with high rate of consanguinity, dual molecular diagnosis is common. Therefore, WES is more accurate method compared to targeted screening to rule out the involvement of modifier genes. It would help in predicting the disease phenotypes. Moreover, bioinformatics analysis was performed to predict the pathogenic effect of the identified variant and to determine the effect of variant on the stability of the mutant CYB5R3 protein.

Comment 3:

The Introduction could be more concise while the discussion would benefit from being more focused on Type I congenital methemoglobinemia as it is relevant to the findings of the investigation. There is also an issue with formatting of the text as part of the results and the Discussion has merged with Figure 2 legend

Answer: Thanks for highlighting. We agree with the worthy reviewer and have concised the introduction section while focusing only on RCM1 in the discussion part. We have removed the citations or studies about the RCM II and have expanded on Type I RCM in the discussion. We regret and apologize for the formatting issue and have rectified it in the revised manuscript.

Comment 4:

Introduction lines 75 to 77 the authors state that the Type II RCM variants affect the FAD binding domain while Type I RCM are associated with the NAD binding domain and refer to reference 13. Figure 3 from this reference shows that both types of variants are present in both regions of the protein. This is too simplified and is related more to the combination of the mutations, whether they cause a truncated protein etc resulting in the total activity of cytochrome b5 reductase.

Answer: We appreciate the reviewer pointing it out. We agree that variants in both the NAD and FAD domains of CYB5R3 have been involved in Type I and Type II RCM. We have rectified the mistake in the introduction. The revision has been highlighted in blue.

Comment 5

Extensive variant classification was performed and presented in parts D and E of Figure 1. The resolution makes it difficult re read. If space is limited then place in a supplementary section so the results of various prediction tools can be analyzed.

Answer: Thanks for mentioning it. We acknowledge the comment and have removed the variant classification from the Figure 1 (section D and E) and have, instead, added it in the Supplementary figure (Table 1).

Comment 6

The term ACMG criteria requires explanation. What criteria did the authors use to define the variant as likely pathogenic? List the criteria used for the variant classification

Answer: By using the ACMG classification, the variant detected in our study is likely pathogenic according to the following criteria.  (PM1: Located in a mutational hot spot and/or critical and well-established functional domain (e.g., active site of an enzyme) without benign variation), PM2: Absent from controls (or a extremely low frequency if recessive) in Exome Sequencing Project, 1000 Genomes Project, or Exome Aggregation Consortium, PP3: Multiple lines of computational evidence support a deleterious effect on the gene or gene product (conservation, evolutionary, splicing impact, etc.) and PP4 (Patient's phenotype or family history is highly specific for a disease with a single genetic etiology) respectively. We have added the criteria in the revised manuscript.

Comment 6:

There are errors of grammar and syntax. The paper would benefit from review by a person whose first language is English and who is familiar with the terminology.

Answer: Thanks for your supportive comments and kind suggestion. We have invited a native English speaker to correct grammatical mistakes and typos and improve the readability of the revised manuscript.

Reviewer 2 Report (New Reviewer)

In the manuscript, it was reported a novel homozygous missense variant in the CYB5R3 gene 380 in a family of Pakistani origin segregating RCM1 in an autosomal recessive manner. It has important medical values which will be helpful in genetic counseling, carrier testing, and bringing awareness to the population. My suggestion is minor revision.

1. The introduction was much simple. Add some research progress and content related to Methemoglobinemia will be better.

2. In the conclusion section, the highlights of the study were not clearly presented and some content (such as the second paragraph of this section) should be included in the discussion section.

3. Please check for spelling and formatting errors throughout the manuscript.

For example:

Line 284,” 4. Discussion” should fall on the next line.

Line 295-296, the font format should be revised.

Author Response

Response to Reviewer 2

Response: We appreciate the reviewing efforts by the reviewer; the proposed changes are very much practical. We have made the changes according to the reviewer’s comments in the manuscript to make things clear.

Comment 1:

  1. The introduction was much simple. Add some research progress and content related to Methemoglobinemia will be better.

Answer: Thanks for your constructive advice. We have looked the literature for more information on methemoglobinemia and have added it in red (introduction section) in the revised manuscript.

Comment 2:

. In the conclusion section, the highlights of the study were not clearly presented and some content (such as the second paragraph of this section) should be included in the discussion section.

Response:

We appreciate the reviewer for pointing this out. We have now added the conclusion in a more suitable and refined manner as marked in red in the revised manuscript. Similarly, as per the worthy reviewer’s suggestion, we have moved the second paragraph of conclusion to the discussion section in the modified manuscript.

Comment 3:

Please check for spelling and formatting errors throughout the manuscript.

Response: Thanks for pointing this out. We have corrected it in the revised manuscript.

Round 2

Reviewer 1 Report (New Reviewer)

Line 66 – replace “can’t” with “is unable to”

Lines 247-249 - Hb Ms are a specific group of Hb variants that have a perturbed heme pocket that allows the iron moiety to be oxidised. Do the authors mean methaemoglobin (MetHb)? Please amend

Table 1 – Amend the term Hb M to MetHb

Line 267 – Please define the criteria as stated in Comment 6 of the covering letter

Line 360 - Amend the term Hb M to MetHb

Author Response

Reviewer 1 (Round 2)

Thank you for your efforts in reviewing this manuscript. Upon your worthy suggestions, we have further improved the quality of this manuscript and have corrected the minor mistakes. We are happy to write that we have addressed your concerns below. We look forward to receiving a positive response from you.

Comment 1

Line 66 – replace “can’t” with “is unable to”

Response: Thanks for pointing it out. We have changed it in the revised manuscript on line 66 as denoted by blue color.

Comment 2

Lines 247-249 - Hb Ms are a specific group of Hb variants that have a perturbed heme pocket that allows the iron moiety to be oxidized. Do the authors mean methemoglobin (MetHb)? Please amend

Response: Thanks for the correction.  We have amended it in the revised manuscript on line 71, 229, 230, 248, 249 and 360 and 368 as denoted by blue color.

Comment 3

Table 1 – Amend the term Hb M to MetHb

Response: Thanks. We have amended it throughout the manuscript.

Comment 4

Line 267 – Please define the criteria as stated in Comment 6 of the covering letter

Response: Thanks for pointing it out. We have defined the ACMG criteria in the main text.

Comment 5

Line 360 - Amend the term Hb M to MetHb

Response: Thanks. We have amended it throughout the manuscript.

This manuscript is a resubmission of an earlier submission. The following is a list of the peer review reports and author responses from that submission.

Round 1

Reviewer 1 Report

Basit et al., have performed MD analysis to explain Methemoglobinemia in patients with mutated CYPB5R3 i.e., reductase of Hb.  The topic is highly important and the authors have attempted to do a good study and analysis.  Nonetheless, there are some major concerns and improvements required before I can recommend publication.  The authors may find the following comments and suggestions useful.

1)      The introduction only gives brief overview about biochemical and clinical data.  Literature on the modelling of the associated proteins and the insights that they provide can be added to give a brief overview to the reader.

2)      The abbreviation RCM should be mentioned at the first use in the introduction (page 2).

3)      On page 3 section 2.5, authors mention “Various in-silico tools were used to determine the pathogenicity …” As per my understanding in-silico tools can be used to predict not determine.  Thus this sentence should be modified.

4)      Section 2.6. “Unfortunately, the crystal structure of CYB5R3 protease has not been resolved to date”, it should be protein, not protease!

5)      Detailed results of the homology modelling should be given in the supporting information.

6)      Section 2.7, contain no information on how the heme of the CYB5R3 was parameterized?  The results are unreliable if this step was skipped, if not then details should be mentioned to assess whether the calculation setup was done correctly or not.

7)      Section 3.4 “Dynamics stability” Here MD simulations are well-known to be stochastic and thus just one simulation of 150 ns may not sample all the possible states for both WT and mutant.  Thus, these simulations are not sufficient to conclude that the mutant is less stable than the WT.  At least a triplicate or 5-repeats for each simulation must be performed and an proper statistical analysis i.e., average, max, min, stdev and p values should be reported.  If computing power is a limiting factor, then, simulation time can be reduced to 100 ns.  Similar argument applies to section 3.5 and 3.6.

8)      Moreover, section 3.5 authors wrote “while 249 p.(Ile224Phe) exhibited a larger fluctuation ranging from 1 to 4.2 Å, consistent with the 250 RMSD curve and conforms to the unstable nature of mutant system”.  It should be noted that MD simulations do not conform anything, they only give valuable insights into the problem one wants to study.  This type of conformation would require time-lapsed X-ray or NMR or Mass-spec studies with a labelled protein.

9)      The authors did not mention/explain how the mutation could influence interaction with Hb and its reduction via the influence on the electron transfer properties of CYB5R3.  These factors are a critical determinant of the conversion of MetHb to Hb.

10)   On page 9 discussion the authors wrote “, one of the patients in our family complained chest pain radiating to left arm”.  It should be “…the patients in the family studied in this work…”

11)   Further the authors mentioned the loss of interactions between p.(Ile224Phe) and NADH domains causes  instability of the CYB5R3 protein, but no data and thorough analysis is present in the results section regarding the same.  Thus it is not clear how they arrived at this conclusion.

12)      No citations are given for the tools used to make predictions discussed in Molecular Genetic Analysis section.

Author Response

Thank you for your efforts in reviewing this manuscript for us. Your comments are very much practical and applicable and will improve the quality of this research. I am happy to write that we have addressed most of your concerns below, and after reviewing the results and the current manuscript, we look forward to receiving a positive response from you.

  • The introduction only gives brief overview about biochemical and clinical data.  Literature on the modelling of the associated proteins and the insights that they provide can be added to give a brief overview to the reader.

Answer: Thanks for pointing it out. We have searched the literature and added additional information on modeling of the proteins in the revised manuscript.

  • The abbreviation RCM should be mentioned at the first use in the introduction (page 2).

Answer: Thanks for mentioning it. We have mentioned RCM as recessive congenital methemoglobinemia at first use in abstract and introduction.

  • On page 3 section 2.5, authors mention “Various in-silico tools were used to determine the pathogenicity …” As per my understanding in-silico tools can be used to predict not determine.  Thus this sentence should be modified.

Answer: Thanks for highlighting. We have rectified it as “predicted” in the revised manuscript.

  • Section 2.6. “Unfortunately, the crystal structure of CYB5R3 proteasehas not been resolved to date”, it should be protein, not protease!

Answer: We appreciate the reviewer pointing it out. We have corrected it as CYB5R3 “protein” in the manuscript.

  • Detailed results of the homology modelling should be given in the supporting information.

Response: Thanks for mentioning the homology modeling. We agree to the worthy reviewer’s suggestion to include the homology modeling with detailed information. We have revised the whole modeling of CYB5R3 and have added more information given in the supplementary file 1.

  • Section 2.7, contain no information on how the heme of the CYB5R3 was parameterized?  The results are unreliable if this step was skipped, if not then details should be mentioned to assess whether the calculation setup was done correctly or not.

Answer: For parameterizing, we need to perform modeling from the start. It is taking too much time. It might be due to low computational power available at our research center.

  • Section 3.4 “Dynamics stability” Here MD simulations are well-known to be stochastic and thus just one simulation of 150 ns may not sample all the possible states for both WT and mutant.  Thus, these simulations are not sufficient to conclude that the mutant is less stable than the WT.  At least a triplicate or 5-repeats for each simulation must be performed and an proper statistical analysis i.e., average, max, min, stdev and p values should be reported.  If computing power is a limiting factor, then, simulation time can be reduced to 100 ns.  Similar argument applies to section 3.5 and 3.6.

Answer: Thanks for giving valuable insight regarding Molecular dynamics simulations. We agree with the reviewer that Molecular dynamics simulation is a stochastic phenomenon and only one simulation of 150 ns may not be sufficient for both WT and MT states. To follow on the reviewer’s suggestion, and to avoid random conclusions, we have run our simulations in triplicate for both the WT and MT with statistical analysis such as average, max, min, stdev values. All the changes are highlighted in the revised manuscript and statistical analysis given in Supplementary File 1 (Table1, 2 and 3) respectively.

  • Moreover, section 3.5 authors wrote “while 249 p. (Ile224Phe) exhibited a larger fluctuation ranging from 1 to 4.2 Å, consistent with the 250 RMSD curve and conforms to the unstable nature of mutant system”.  It should be noted that MD simulations do not conform anything, they only give valuable insights into the problem one wants to study.  This type of conformation would require time-lapsed X-ray or NMR or Mass-spec studies with a labelled protein.

Answer: Thanks for pointing out the mistake. We agree with the reviewer that MD simulations don’t conform anything but just provide molecular insights into the dynamic properties of proteins. We have removed the “conformation” where it was written and have given statistical analysis for the MD simulation run in triplicate just to provide an insight regarding the behaviours of CYB5R3 wild and mutant systems.

  • The authors did not mention/explain how the mutation could influence interaction with Hb and its reduction via the influence on the electron transfer properties of CYB5R3.  These factors are a critical determinant of the conversion of MetHb to Hb.

Answer: Thanks for mentioning it. We acknowledge the comment and have explained briefly the interaction of Hb.

  • On page 9 discussion the authors wrote “, one of the patients in our family complained chest pain radiating to left arm”.  It should be “…the patients in the family studied in this work…”

Answer: Thanks for pointing it out. We have corrected it to “the patients in the family studied in this work”

  • Further the authors mentioned the loss of interactions between p.(Ile224Phe) and NADH domains causes  instability of the CYB5R3 protein, but no data and thorough analysis is present in the results section regarding the same.  Thus it is not clear how they arrived at this conclusion.

Answer: We are sorry for the mistake and we have corrected it in the revised manuscript.

  • No citations are given for the tools used to make predictions discussed in Molecular Genetic Analysis section.

Thanks for pointing it out. We have added the citations for the tools used for prediction of the missense variant in the revised manuscript.

Reviewer 2 Report

Molecular Dynamic Simulation Analysis of a Novel Missense Variant in CYB5R3 Gene in Patients with Methemoglobinemia:

Reviewer comments:

1] Page 1, Line No.17.  Mutations in CYB5R3 gene cause reduced or absent NADH-dependent cytochrome b5 17 reductase enzyme function and consequently leads to recessive congenital methemoglobinemia 18 (RCM).

Mutations in CYB5R3 gene could not cause the absence of NADH-dependent cytochrome b5  reductase enzyme activity/ function.

2] How did the author measure the HbM level?  (Detail method is not given).

3] Page No. 4 , Line No. 183…. On further examination, her hemoglobin studies  showed bands of HbA, HbA2, and HbM. If Hb M (49%) variant is present, then how could the NADH-cytochrome b5 reductase enzyme deficiency also be present?

4] Author should perform the NADH-cytochrome b5 reductase enzyme activity assay.

5]   Page No. 1, Line No. 27 ---à Data analysis of WES revealed a novel homozygous missense variant 27 NM_001171660.2:c.670A>T: NP_001165131.1:p.(Ile224Phe) in exon 7 of the CYB5R3 gene located on 28 chromosome 22q13.2.

Please confirm the position of this novel variant,  it should be in exon 8 but not in exon 7.

6] Family pedigree is  not properly drawn. Since both the parents are heterozygous for this variant p.(Ile224Phe),  heterozygosity/ carrier representation is absent in the family pedigree.

7] The author has not done any functional study for this novel variant p.(Ile224Phe) . They used only bioinformatics tools to predict the impact of this novel variant on protein structure and functions.

Author Response

Response to Reviewer 2

Response: We appreciate the reviewing efforts by the reviewer, the proposed changes are very much practical. We have made the changes according to the reviewer’s comments in the manuscript to make things clear.

Reviewer comments:

1] Page 1, Line No.17.  Mutations in CYB5R3 gene cause reduced or absent NADH-dependent cytochrome b5 17 reductase enzyme function and consequently leads to recessive congenital methemoglobinemia 18 (RCM). Mutations in CYB5R3 gene could not cause the absence of NADH-dependent cytochrome b5 reductase enzyme activity/ function.

Answer: Thanks for pointing out this mistake. We apologize for the mistake and we have corrected it to “decreased function of cyb5r3” in the revised manuscript.

2] How did the author measure the HbM level?  (Detail method is not given).

Answer: The following lines have been added in the revised manuscript:

The diagnosis is confirmed by direct measurement of methemoglobin by a multiple wavelength co oximeter. On a blood gas, normal PO2 concentrations are usually found on analysis. Clinical Cyanosis in the presence of normal arterial oxygen tensions is highly suggestive of methemoglobinemia. Pedigree was drawn after obtaining the in-formation from elder members of the family.

3] Page No. 4, Line No. 183…. On further examination, her hemoglobin studies showed bands of HbA, HbA2, and HbM. If Hb M (49%) variant is present, then how could the NADH-cytochrome b5 reductase enzyme deficiency also be present?

Answer: We appreciate the reviewer for pointing out this glaring mistake. As per the worthy reviewer’s highlighting we have corrected it and have removed “the deficiency” of his enzyme in the modified manuscript.

4] Author should perform the NADH-cytochrome b5 reductase enzyme activity assay.

Answer: We requested the patients for samples to perform the NADH-cytochrome b5 reductase enzyme activity assay, but unfortunately, they declined to provide samples.

5]   Page No. 1, Line No. 27 ---à Data analysis of WES revealed a novel homozygous missense variant 27 NM_001171660.2:c.670A>T: NP_001165131.1:p.(Ile224Phe) in exon 7 of the CYB5R3 gene located on 28 chromosome 22q13.2. Please confirm the position of this novel variant, it should be in exon 8 but not in exon 7.

   Answer:  We apologize for the mistake and have corrected it. Thanks for highlighting it.

6] Family pedigree is not properly drawn. Since both the parents are heterozygous for this variant p.(Ile224Phe),  heterozygosity/ carrier representation is absent in the family pedigree.

Thanks for pointing it out. We acknowledge the issue and have modified our pedigree in the manuscript.

7] The author has not done any functional study for this novel variant p.(Ile224Phe) . They used only bioinformatics tools to predict the impact of this novel variant on protein structure and functions.

Answer: We are thankful to the reviewer for reviewing our manuscript. As mentioned we carried bioinformatic analyses of our identified missense variant in CYB5R3. Unfortunately, we could not perform the functional study of this variant.

Reviewer 3 Report

Ullah and coauthors report a novel variant of Cyb5R3 with an Ile224Phe mutation in exon 7 in a consanguinous Pakistani family. Whole exome sequencing and Sanger sequencing were used for identifying the causative gene in the disease phenotype presented by the participating family members (n=7). ACMG guidelines and bioinformatic tools were used to determine the pathogenicity of the variant protein with NADH domain aberrant function detected by studying molecular dynamics.  The authors suggest their findings will “facilitate genetic counselling” in families carrying mutations in the CYB5R3 gene and even suggest that their findings “will assist in the prognosis and prevention of congenital disorders”.  

Overall, the information in the manuscript leaves the reader with many questions, a large number of which pertain to how to interpret what the authors are trying to say. This is due to numerous grammatical errors, which could have been detected by a spelling check before submission.  In particular, there is no information on the frequency of the new CYB5R3 variant in the general Pakistani population and it’s likely the variant does not occur much outside the remote region in which the family under investigation resides.  The relevance of this new variant to the national or global cases of CYB5R3-related methemoglobinemia will likely be small. Hence, the manuscript needs considerable rewriting and is of low impact.

1- The manuscript does not state an objective or hypothesis for the undertaken studies.

2- A specific consanguineous family residing in a “remote region of the country” was used to research the CYB5R3 variant. The two patients/subjects used to describe the clinical phenotype of the variant are closely related (cousins).  The authors neither explore the limitations of their findings given that non-CYB5R3 artifact may be contributed by consanguineous marriage, nor do they compare/contrast the clinical phenotypes of carriers versus noncarriers of the Cyb5R3 variant. There were 7 participating family members in the study from 2 different generations but only data for 2 carriers of the variant are provided when more could have been made available.

3- Nonstandard English grammar is used throughout the manuscript, making it difficult to understand and read. Many sentences lack the requisite definite and indefinite articles, noun/verb agreement and proper tense.  The reader is left with a general sense of carelessness in the construction of the manuscript, both text and figures. The number of errors in the manuscript are too numerous to list them all.  The following list is a sampling.

Line 74: “Family” should not be capitalized

Line 64: “complex interlay” is likely misspelled

Lines 102-103: Sentence is incomplete and incomprehensible

Line 106-107: Sentence is grammatically incorrect. The definite article “the” should precede “polymorphic nature”

There are too many instances of missing definite (the) and indefinite (a/an) articles to list them all (lines 168-169, 176, 237, 241, 291, 296, 309, 312, 314, 323, 329, 334, 335, 339, 348)

Lines 123-125: an explanation of these “in silico tools” would benefit the reader who is not familiar with them.

Line 177: should be “erythrocyte morphology”, singular not plural

Table 1: The number of thrombocytes for patient 1 is listed as “24 x 109”, however in the text it is written as “241 x 109” (line 179).  “Meth-Hb” is misspelled; it should be metHb.

Line 223: “an in” should be “and in”

Line 232: should be “and disrupt its normal activity”

Line 238: “give rise” should be “gave rise” – wrong tense

Line 251: “and conforms” does not make sense

Line 251-253: requires more punctuation, like commas

Line 249: what is the significance of the statement regarding a “converged Rg”.  How should the reader interpret this?

Line 280: “have cyanosis” should be changed to “positive for cyanosis”

Line 289: “Till date” is awkward; it should be replaced with “Until now”

Line 294: “originated” should be “originating”

Line 296: “sever” should be “severe”

Line 312: should be “complained of chest pain”

Line 316: noun and verb are in disagreement. Should be “were unremarkable”

Lines 322-324: punctuation is needed (commas)

Line 327: “details” should be singular – “detail”

Line 330-333: Sentence does not make sense as written

Line 335- 337: Sentence does not make sense as written

Line 337-339: Sentence requires more punctuation (comma)

4- Figure legends are not complete, failing to define symbols used in figures, such as circles and squares in panel A of Figure 1.

5- The authors claim in the abstract that “The present study expanded the phenotypic spectrum resulting from a novel sequence variant of the CYB5R3.”  The authors do not indicate which findings lead to that conclusion. They also contend that their findings will “…assist in the prognosis and prevention of congenital disorders” but they do not give examples of how this could be achieved.

6- There are underlined sentences/fragments for unknown reasons:  line 52, lines 279-283.

7- The authors claim that the Cyb5R3 crystal structure has not been resolved to date (Lines 133-134. This is inaccurate.  Various isoforms of the protein (minus the membrane-anchoring segment) can be found online using NCBI (https://www.ncbi.nlm.nih.gov/structure/?term=cyb5r3). Secondly, they refer to CYB5R3 as a “protease”, which it is not.  It is a protein with reductase activity.

Author Response

Response to Reviewer 3

Ullah and coauthors report a novel variant of Cyb5R3 with an Ile224Phe mutation in exon 7 in a consanguineous Pakistani family. Whole exome sequencing and Sanger sequencing were used for identifying the causative gene in the disease phenotype presented by the participating family members (n=7). ACMG guidelines and bioinformatic tools were used to determine the pathogenicity of the variant protein with NADH domain aberrant function detected by studying molecular dynamics.  The authors suggest their findings will “facilitate genetic counselling” in families carrying mutations in the CYB5R3 gene and even suggest that their findings “will assist in the prognosis and prevention of congenital disorders”.  

Comment: Overall, the information in the manuscript leaves the reader with many questions, a large number of which pertain to how to interpret what the authors are trying to say. This is due to numerous grammatical errors, which could have been detected by a spelling check before submission.  In particular, there is no information on the frequency of the new CYB5R3 variant in the general Pakistani population and it’s likely the variant does not occur much outside the remote region in which the family under investigation resides.  The relevance of this new variant to the national or global cases of CYB5R3-related methemoglobinemia will likely be small. Hence, the manuscript needs considerable rewriting and is of low impact.

Response: We are very grateful to the reviewer for the comprehensive review of our manuscript. The comments by the reviewer are practical that will significantly improve our paper. We have been able to incorporate changes to reflect most of the suggestions provided by the reviewer. Acknowledging this fundamental limitation of our English language, we have revised the manuscript again and have corrected the words/sentences where necessary. The corrected words/sentences are highlighted in blue in the revised manuscript.

Regarding the frequency of the identified CY5R3 variant, The variant detected is absent from population databases such as gnomAD, 1000 Genomes, ESP6500 and 183 ethnically matched control exomes. We have added these information in the manuscript.

Comment: The manuscript does not state an objective or hypothesis for the undertaken studies.

Response: We have incorporated objective of the study in the revised manuscript.

Comment:   2- A specific consanguineous family residing in a “remote region of the country” was used to research the CYB5R3 variant. The two patients/subjects used to describe the clinical phenotype of the variant are closely related (cousins).  The authors neither explore the limitations of their findings given that non-CYB5R3 artifact may be contributed by consanguineous marriage, nor do they compare/contrast the clinical phenotypes of carriers versus noncarriers of the Cyb5R3 variant. There were 7 participating family members in the study from 2 different generations but only data for 2 carriers of the variant are provided when more could have been made available.

Answer: Methemoglobinemia is a recessive disorder caused by homozygous/biallelic variants in CYB5R3. We performed whole exome sequencing in two patients and found a pathogenic sequence variant in CYB5R3. The findings were validated by Sanger sequencing in the DNA of affected and normal individuals (parents and siblings of the patients). Individuals who were homozygous for the identified CYB5R3 variant were affected with methemoglobinemia, while those who were heterozygous or wild type were healthy. In addition to 7 members of the present study we tested exomes of 183 healthy controls but the variant was not found outside the family under study. We have included these information in the manuscript.

Comment 3- Nonstandard English grammar is used throughout the manuscript, making it difficult to understand and read. Many sentences lack the requisite definite and indefinite articles, noun/verb agreement and proper tense.  The reader is left with a general sense of carelessness in the construction of the manuscript, both text and figures. The number of errors in the manuscript are too numerous to list them all.  The following list is a sampling.

Line 74: “Family” should not be capitalized

Line 64: “complex interlay” is likely misspelled

Lines 102-103: Sentence is incomplete and incomprehensible

Line 106-107: Sentence is grammatically incorrect. The definite article “the” should precede “polymorphic nature”

There are too many instances of missing definite (the) and indefinite (a/an) articles to list them all (lines 168-169, 176, 237, 241, 291, 296, 309, 312, 314, 323, 329, 334, 335, 339, 348)

Lines 123-125: an explanation of these “in silico tools” would benefit the reader who is not familiar with them.

Line 177: should be “erythrocyte morphology”, singular not plural

Table 1: The number of thrombocytes for patient 1 is listed as “24 x 109”, however in the text it is written as “241 x 109” (line 179).  “Meth-Hb” is misspelled; it should be metHb.

Line 223: “an in” should be “and in”

Line 232: should be “and disrupt its normal activity”

Line 238: “give rise” should be “gave rise” – wrong tense

Line 251: “and conforms” does not make sense

Line 251-253: requires more punctuation, like commas

Line 249: what is the significance of the statement regarding a “converged Rg”.  How should the reader interpret this?

Line 280: “have cyanosis” should be changed to “positive for cyanosis”

Line 289: “Till date” is awkward; it should be replaced with “Until now”

Line 294: “originated” should be “originating”

Line 296: “sever” should be “severe”

Line 312: should be “complained of chest pain”

Line 316: noun and verb are in disagreement. Should be “were unremarkable”

Lines 322-324: punctuation is needed (commas)

Line 327: “details” should be singular – “detail”

Line 330-333: Sentence does not make sense as written

Line 335- 337: Sentence does not make sense as written

Line 337-339: Sentence requires more punctuation (comma).

Response: We are grateful for the detail pointing out of mistakes in our manuscript. We believe that rectifying these mistakes will significantly improve the quality of our manuscript. We have incorporated significant changes in the manuscript and have further rectified all the mistakes as pointed out by the worthy reviewer in the revised manuscript. The corrected words/sentences are highlighted in blue.

4- Figure legends are not complete, failing to define symbols used in figures, such as circles and squares in panel A of Figure 1.

Answer: We very much appreciate this helpful comment. We have added complete figure legends and have clarified the symbols used in Figure 1.

5- The authors claim in the abstract that “The present study expanded the phenotypic spectrum resulting from a novel sequence variant of the CYB5R3.”  The authors do not indicate which findings lead to that conclusion. They also contend that their findings will “…assist in the prognosis and prevention of congenital disorders” but they do not give examples of how this could be achieved.

Answer: The present findings of the study would be help in clinical genetics where the information can be used for genetic counseling, prenatal diagnosis and carrier testing. For example, if the individuals who are heterozygous carrier of the identified CYB5R3 get consanguineous marriage, they have higher probability to segregate the mutated alleles to their offspring in homozygous state leading to disease phenotypes. Carrier testing for the CYB5R3 variant could help these individuals to know their genotypes and to make decision for avoiding cousin marriage to prevent the disease in their upcoming generations.

Similarly, the findings of the study can help parents of the patients to know about the genotypes of next fetus by prenatal testing.

6- There are underlined sentences/fragments for unknown reasons:  line 52, lines 279-283.

 Response: We have removed the underlined sentences. Thanks for pointing it out.

7- The authors claim that the Cyb5R3 crystal structure has not been resolved to date (Lines 133-134. This is inaccurate.  Various isoforms of the protein (minus the membrane-anchoring segment) can be found online using NCBI (https://www.ncbi.nlm.nih.gov/structure/?term=cyb5r3). Secondly, they refer to CYB5R3 as a “protease”, which it is not.  It is a protein with reductase activity.

Answer: Thanks for highlighting it. We agree with the reviewer that Cyb5R3 crystal structure has not been resolved to date, is incorrect. We apologize for the mistake and we corrected it accordingly. Additionally, we also agree that CYB5R3 is not a “protease” but a protein with reductase activity. We have corrected it to “protein” in the revised manuscript.

Round 2

Reviewer 1 Report

The authors addressed some of the minor comments. Nonetheless, they did not explain how MD simulations were set up and how heme was parameterized.  This is not given in the SI also. Thus the results are not reliable. Within amber software, there are multiple ways of parameterizing metal binding site. None of those were used, thus the prmtop/inpcrd input files for MD are unlikely to have contained there parameters and thus the simulations are unreliable. 

Reviewer 2 Report

No satisfactory explanation

Reviewer 3 Report

1- The authors do not adequately address the following important concern:

"The authors neither explore the limitations of their findings given that non-CYB5R3 artifact may be contributed by consanguineous marriage, nor do they compare/contrast the clinical phenotypes of carriers versus noncarriers of the Cyb5R3 variant. There were 7 participating family members in the study from 2 different generations but only data for 2 carriers of the variant are provided when more could have been made available."

Simply stating that family members without homozygous expression of the mutant Cyb5R3 are "healthy" is not sufficient.

2- The English language is still incorrectly applied in many instances.  A particular case is the addition at lines 391-394, which is incomprehensible as written: "It should be noted that, MD simulations, though are helpful in giving prediction about genetic variants in disease causation, still, it is an abstract representation to model human diseases, and are limited by in vitro and in vivo techniques."